# Diagnostic role of heart rate variability in breast cancer and its relationship with peripheral serum carcinoembryonic antigen

Lishan Ding[1]◉, Yuepeng Yang[1]◉, Mingsi Chi[1]◉, Zijun Chen[1‡], Yaping Huang[1‡], Wenshan Ouyang[1‡], Weijian Li[1], Lei He[2]*, Ting Wei[3]*

1 The Second Clinical College of Southern Medical University, Guangzhou, Guangdong, China,
2 Department of Endocrinology, Zhujiang Hospital, Southern Medical University, Guangzhou, Guangdong, China, 3 Department of Oncology, Zhujiang Hospital, Southern Medical University, Guangzhou, Guangdong, China

◉ These authors contributed equally to this work.
‡ ZC, YH and WO also contributed equally to this work.
* 765139701@qq.com (LH); 258077534@qq.com (TW)

**Data Availability Statement:** All relevant data are within the manuscript and its Supporting information files.

## Abstract

### Objective

To investigate the diagnostic role of heart rate variability in breast cancer and its relationship with Carcinoembryonic antigen (CEA) in peripheral serum.

### Methods

We reviewed the electronic medical records of patients who attended Zhujiang Hospital of Southern Medical University between October 2016 and May 2019. The patients were grouped based on breast cancer history and were divided into two groups: breast cancer group(n = 19) and control group(n = 18). All women were invited for risk factor screening, including 24-hour ambulatory ECG monitoring and blood biochemistry after admission. The difference and correlation between the breast cancer group and control group were performed by comparing the heart rate variability and serum CEA levels. Additionally, diagnostic efficacy analysis of breast cancer was calculated by combining heart rate variability and serum CEA.

### Results

In total, 37 patients were eligible for analysis, with 19 and 18 patients in the breast cancer group and control groups, respectively. Women with breast cancer had a significantly lower level of total LF, awake TP, and awake LF, and a significantly higher level of serum CEA compared with women with no breast cancer. Total LF, awake TP, and awake LF were negatively correlated with the CEA index (P < 0.05). The receiver operating characteristic (ROC) curves indicated the highest area under the curve (AUC) scores and specificity of the combination of awake TP, awake LF, and serum CEA (P < 0.05), while sensitivity was highest for total LF, awake TP, and awake LF (P < 0.05).

**Funding:** The authors received no specific funding for this work.

**Competing interests:** The authors have declared that no competing interests exist.

## Conclusions

Women with history of breast cancer had abnormalities in autonomic function. The combined analysis of heart rate variability and serum CEA analysis may have a predictive effect on the development of breast cancer and provide more basis for clinical diagnosis and treatment.

## Introduction

Breast cancer is caused by a variety of carcinogens, leading to the uncontrolled proliferation of breast epithelial cells. According to the Global Cancer Update provided by Global Cancer Statistics 2020, there were 226,419 new confirmed cases of breast cancer and 684,996 new deaths. Breast cancer has surpassed lung cancer as the most common cancer in women and one of the highest incidences and mortality in women [1, 2]. Treatment options for breast cancer include targeted therapy, endocrine therapy, radiation therapy, surgery, and chemotherapy [3]. Clinically, the most suitable treatment for breast cancer patients is determined based on tumor subtype and cancer stage [4]. For example, novel treatment options, including targeted therapy and immunotherapy, have emerged in recent years for metastatic triple negative breast cancer [5–7]. Breast-conserving surgery is often used for patients with early-stage breast cancer, while mastectomy is considered the most effective method for patients with advanced breast cancer [8]. However, breast cancer patients often lose obvious symptoms in their early stages which are already in the intermediate and advanced stages, and miss the best time for treatment [9], with a low survival rate and easy recurrence [10–12].

Currently, common diagnostic methods for breast cancer include mammography, ultrasound scan, fine needle aspiration, and tumor marker testing (e.g., CA199, CEA, CA15-3, CA125) [13, 14]. Compared with other diagnostic methods, tumor marker testing is quick and easy, basically non-invasive and cost-effective in early cancer diagnosis leading to a better reflection of tumor development and the body's response to the tumor. Carcinoembryonic antigen (CEA) is mainly used for clinical monitoring of colorectal cancer, gastric cancer, breast cancer, pancreatic cancer, hepatocellular carcinoma, lung cancer, and medullary thyroid cancer, which is of great value in the diagnosis, screening, and prognosis of tumors [15]. Recent studies have shown that preoperative CEA levels may provide useful for the identification and treatment of breast cancer [16]. Wu et al [17]. showed that serum CEA levels were elevated during breast cancer. And the European Tumor Markers Panel recommended CEA levels as an indicator for the assessment of prognosis, early detection of disease progression, and treatment monitoring in breast cancer patients [18]. However, the specificity of CEA for early diagnosis of breast cancer is relatively low. Therefore, we sought to combine other diagnostic methods to improve the efficacy.

Heart rate variability (HRV) refers to the change between each cardiac cycle, which originates from the autonomic regulation of the heart's sinus node. It is considered to be an important indicator of autonomic function and action, reflecting the balance between the vagus and sympathetic nerves. Studies have found that patients with breast cancer have a higher risk of cardiovascular disease which presented a lower HRV, implying vagal dysfunction [19–21]. Karolina Majerova et al. [22] showed cardiac vagal modulation alterations in breast cancer survivors by measuring HRV, revealing a significant increase in sympathetic modulation in breast cancer patients relative to healthy volunteers. In addition, previous studies have shown that HRV analysis can help determine tumor staging, efficacy, prognosis, and autonomic function

[23–25]. In Desmond G. Powe's trial [26], beta-blocker therapy significantly reduced distant metastasis, cancer recurrence, and cancer-specific mortality in breast cancer patients, suggesting that sympathetic inhibition can inhibit breast cancer progression. To be concluded, breast cancer patients have impaired autonomic nervous system activity so early recognition is clinically significant to increase treatment chances and survival time. Therefore, HRV, as a non-invasive measure widely used in clinical practice to assess autonomic nervous system activity [27], may be a clinical tool for detecting early breast cancer.

In summary, it is insidious and lacks of effective screening methods to diagnose in the early stages of breast cancer, which seriously affects the life and health of women. Therefore, this study aimed to investigate the changes in heart rate variability and carcinoembryonic antigen in breast cancer patients and their role in the diagnosis of breast cancer, to provide a new adjunctive method for early diagnosis of breast cancer, and to improve the detection rate [28].

## Materials and methods

The key elements of this study were to select qualified cancer patients by establishing exclusion criteria and select appropriate statistical methods to analyze the data according to the data characteristics. A large number of studies have shown that in addition to cancer, inflammation, cardiovascular disease, metabolic disease, dyslipidemia and other diseases affecting heart rate variability. Therefore, the patients in the cancer group were excluded from the above diseases that might interfere with heart rate variability in this study. Meanwhile, statistical methods were used to compare the various blood lipid indicators of patients in the cancer group and the control group of healthy patients (the difference was not statistically significant when P>0.05). The Ethics Committee of Zhujiang Hospital approved this study (NO.2022-KY-044).

### Participants and procedures

The electronic medical records of patients diagnosed with breast cancer at Zhujiang Hospital of Southern Medical University from October 1, 2016 to May 1, 2019 were selected for retrospective analysis in this study.

Inclusion criteria: 1) patients with complete general clinical information; 2) meeting the diagnostic criteria for breast cancer lesions.

Exclusion criteria: 1) organic heart disease such as heart failure; 2) pre-existing palpitations, abnormal heart rate, etc.; 3) hyperthyroidism; 4) diabetes mellitus; 5) inflammation; 6) infection.

The study finally included 19 cases of breast cancer group, aged 40–78 years, with a mean age of (58.42±12.76) years; another 18 cases of healthy women, aged 32–71 years, with a mean age of (51.11±11.82) years, who were included in the voluntary test and had complete general clinical data at the same time, were selected as the control group. Data were compared between the two groups for baseline characteristics with P > 0.05, and the differences were not statistically significant or comparable.

### Measures

The quantitative variables in this study were not grouped.

**Biochemistry.** Venous blood was collected from all subjects in a 12-h fasting state and the early morning of the following day, and the following venous blood parameters were measured using a Beckman CX5 automatic biochemistry analyzer: Alanine aminotransferase (ALT), aspartate aminotransferase (AST), aspartate aminotransferase/alanine aminotransferase ratio (AST/ALT), urea, total cholesterol (TC), total bilirubin (TBIL), total protein (TP), total calcium (Ca), globulin (GLO), triglycerides (TG), albumin (ALB), albumin/globulin ratio

(ALB/GLO), direct bilirubin (DBIL), alkaline phosphatase (ALP), creatinine (Crea), glucose (Glu), indirect bilirubin (IBIL), carcinoembryonic antigen (CEA), and other items.

**HRV.** All subjects recorded test results using the domestic BIHONKOHDEN ambulatory ECG workstation recorder RAC-3012 and performed heart rate variability analysis of 24h ambulatory ECG using its analysis system. Recorded time domain metrics: 1) total standard deviation of normal sinus RR interval (SDNN); 2) the mean standard deviation of sinus RR interval every 5 minutes (SDNNin); 3) the root mean square of normal continuous sinus RR interval (rMSSD); 4) the percentage difference of adjacent RR interval > 50 ms (pNN50). Recorded frequency domain indicators: 1) total power (TP); 2) very low frequency power (VLF); 3) low frequency power (LF); 4) high frequency power (HF).

## Statistical analysis

Data were analyzed using SPSS 26.0 statistical software, and data normality was tested by the Shapiro-Wilk method. For indicators conforming to the normal distribution, their intergroup comparisons were performed by independent sample t-test, expressed as mean ± standard deviation (x±s); for indicators not conforming to the normal distribution, their intergroup comparisons were performed by rank sum test, expressed as median (interquartile spacing), i.e., M (P25, P75). Spearman's method was used for correlation analysis; binary logistic regression analysis was used for multifactorial analysis to assess the risk relationship in terms of odds ratio (OR) and 95% confidence interval (CI). Assessing collinearity between independent variables using collinearity diagnostics. The goodness of fit was tested by Hosmer-Lemeshow test. The receiver operating characteristic curve (ROC) was used to assess the diagnostic efficacy of CEA and HRV for breast cancer, and the critical value, sensitivity, and specificity were calculated. $P < 0.05$ was considered a statistically significant difference.

## Results

All study data were obtained from 19 women with breast cancer and 18 women in the control group. Participants with missing data were not included in this study.

## Participants selection

Using the methods described above, we identified 229 patients who were diagnosed with breast cancer (Fig 1). In total, 210 of 229 patients were excluded for incomplete data and specific diseases. In details, 179 patients were not available in HRV. Another 14 of 50 patients with HRV were excluded as they did not contained CEA. Seventeen patients were excluded because these patients were diagnosed diseases which may affect heart rate including organic heart disease such as heart failure (1), abnormal heart rate (3), hyperthyroidism (1), diabetes mellitus (1), inflammation or infection (11). Then we identified 562 people who did not suffer from cancer, inflammation, infection, heart failure, diabetes, hyperthyroidism, fever, leukemia, anemia control group during the same period. 343 people were not available in HRV. Another 168 men with HRV were excluded. Then 33 of 51 patients with HRV were excluded because they did not contained CEA. The study finally included 19 cases of breast cancer group and 18 cases of healthy women as the control group. The study finally included 19 cases of breast cancer group and 18 cases of healthy women as the control group.

## Comparison of general information between the two groups

The age, weight, height, body mass index (BMI), total cholesterol (TC), total protein (TP), albumin (ALB), albumin/globulin (ALB/GLO), direct bilirubin (DBIL), and creatinine (Crea)

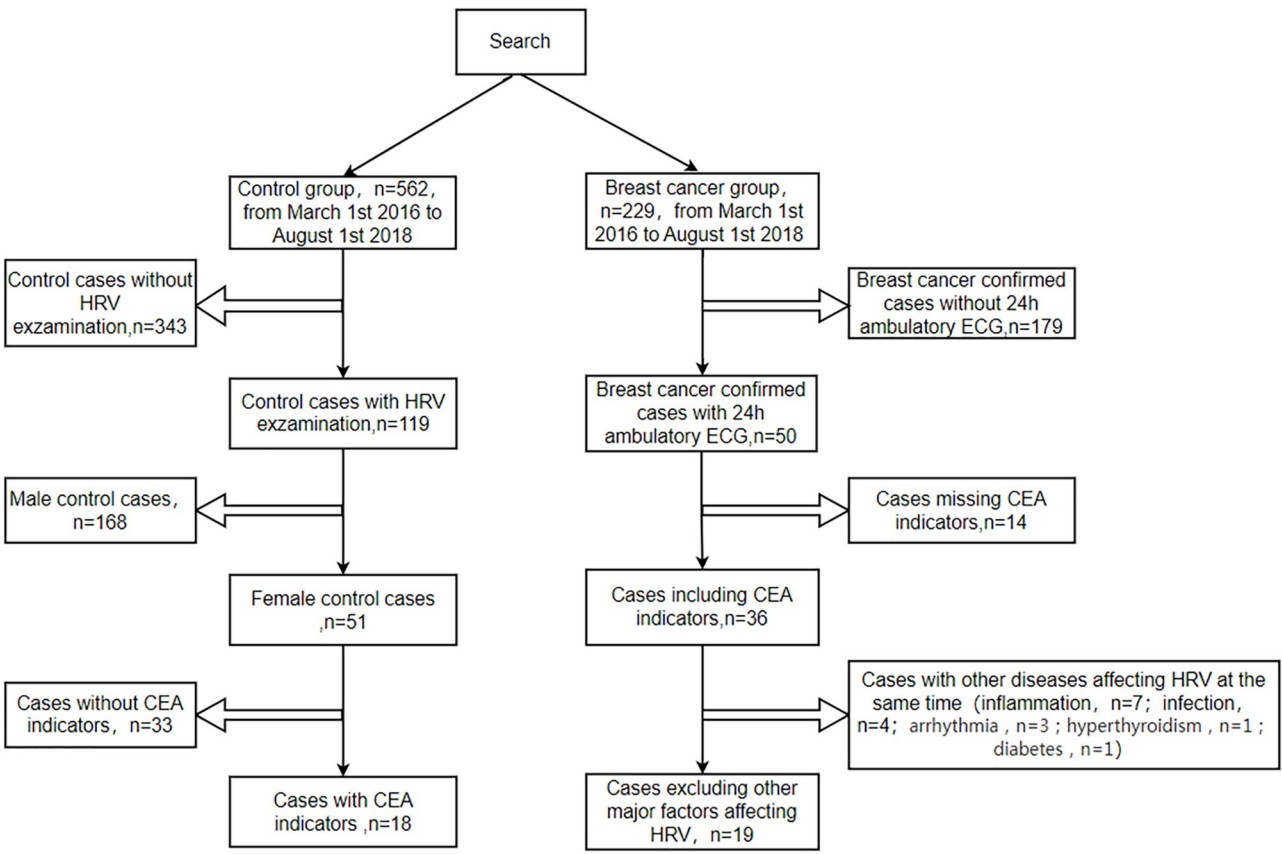

**Fig 1. Inclusion and exclusion criteria for patient selection.** HRV, heart rate variability; CEA, carcinoembryonic antigen; ECG, electrocardiogram.

indexes of the two groups conformed to the normal distribution. An independent samples t-test was conducted to investigate possible differences from baseline between the group's statistics. There was no significant difference from baseline between the groups (P>0.05).

Basal metabolic rate (*BMR% = systolic blood pressure–diastolic blood pressure + pulse count–110*), alanine aminotransferase (ALT), aspartate aminotransferase (AST), aspartate aminotransferase/alanine aminotransferase ratio (AST/ALT), urea, total bilirubin (TBIL), total calcium (Ca), globulin (GLO), triglyceride (TG), alkaline phosphatase (ALP), glucose (Glu), and indirect bilirubin (IBIL) indicators did not conform to the normal distribution. A rank sum test was conducted to verify possible differences from baseline between the group's statistics. There was no significant difference from baseline between the groups (P>0.05) (S1 Table and Table 1).

## Comparison of HRV and CEA indicators between the two groups

The total VLF, awake SDNN, and sleep VLF in both groups conformed to the normal distribution (P>0.05). An independent sample t-test was conducted to investigate possible differences between the group's statistics. And there was no significant difference between the groups (P>0.05).

TP, total LF, total HF, total SDNN, total SDNNin, total rMSSD, total pNN50, awake TP, awake VLF, awake LF, awake HF, awake SDNNin, awake rMSSD, awake pNN50, sleep TP,

**Table 1. Comparison of general information between the two groups.**

| Variables | Control group | Breast cancer group | F /Z-value | P-value |
|---|---|---|---|---|
| Age[a](year) | 51.11±11.82 | 58.42±12.76 | 0.561 | 0.080 |
| Weight[a](kg) | 51.81±8.86 | 57.03±7.84 | 0.339 | 0.066 |
| Height[a](cm) | 157.22±3.12 | 159.21±3.55 | 0.051 | 0.080 |
| BMI[a](kg/m$^2$) | 20.97±3.54 | 22.54±3.31 | 0.091 | 0.170 |
| BMR[b](%) | 14.00(4.50,19.50) | 18.00(10.00,25.00) | -1.234 | 0.217 |
| ALT[b](IU/L) | 11.50(9.00,16.25) | 20.00(10.00,25.00) | -1.906 | 0.057 |
| AST[b](IU/L) | 16.50(13.75,19.25) | 19.00(15.00,25.00) | -1.431 | 0.152 |
| AST/ALT[b] | 1.40(1.08,1.58) | 1.20(1.00,1.60) | -0.688 | 0.492 |
| Urea[b](mmol/L) | 4.57(4.06,5.26) | 4.77(3.44,5.32) | -0.319 | 0.750 |
| TC[a](mmol/L) | 5.15±0.63 | 5.37±0.87 | 1.190 | 0.390 |
| TBIL[b](μmol/L) | 9.62(7.60,11.48) | 8.90(7.40,10.90) | -0.851 | 0.395 |
| TP[a](g/L) | 71.36±4.99 | 68.35±7.91 | 1.820 | 0.179 |
| Ca[b](mmol/L) | 2.29(2.23,2.36) | 2.31(2.26,2.45) | -0.639 | 0.523 |
| GLO[b](g/L) | 29.35(26.68,31.43) | 27.70(23.20,29.10) | -1.520 | 0.129 |
| TG[b](mmol/L) | 0.99(0.75,1.47) | 1.24(0.86,1.51) | -1.155 | 0.248 |
| ALB[a](g/L) | 42.62±3.37 | 10.72±4.51 | 1.045 | 0.158 |
| ALB/GLO[a] | 1.47±0.20 | 1.50±0.22 | 0.179 | 0.691 |
| DBIL[a](μmol/L) | 4.33±2.00 | 4.50±1.30 | 0.953 | 0.764 |
| ALP[b](IU/L) | 60.50(52.50,69.25) | 65.00(53.00,77.00) | -0.714 | 0.475 |
| Crea[a](μmol/L) | 62.65±11.96 | 64.41±19.20 | 1.102 | 0.742 |
| Glu[b](mmol/L) | 5.05(4.60,5.53) | 5.00(4.53,6.40) | -0.030 | 0.976 |
| IBIL[b](μmol/L) | 5.55(3.63,6.83) | 4.50(3.60,6.20) | -0.745 | 0.456 |

[a] All groups of this index followed a normal distribution and were expressed as $\bar{x} \pm s$. Independent samples t-test was used for comparison between groups.

[b] At least one of the groups of this indicator did not follow a normal distribution, denoted by M (P25, P75), and comparisons between groups were made using rank sum test.

sleep LF, sleep HF, sleep SDNN, sleep SDNNin, sleep rMSSD, sleep pNN50 HRV parameters and CEA indicators did not conform to the normal distribution. A rank sum test was conducted to investigate possible differences between the group's statistics.

The comparison of HRV parameters between the control (n = 18) and breast cancer (n = 19) group was shown in the S2 Table and Fig 2. Total LF, awake total power, and awake LF HRV parameters and CEA indicators were significantly lower in the breast cancer group (283.90 (195.50, 554.50) vs 213.40 (80.00, 421.70), 1298.50 (1044.00, 2033.75) vs 816.00 (454.00, 1519.80), 298.50(214.75,612.00) vs 152.00(84.00,344.00), and 1.25(0.70,1.63) vs 2.70 (1.60,5.20) $P$ = 0.045, 0.033, 0.019, and <0.001, respectively). The differences of the remaining indicators were not statistically significant (P>0.05).

## Multifactorial logistic regression analysis of breast cancer group

A logistic regression was performed to ascertain CEA and HRV parameters on the likelihood that participants have breast cancer. In multivariate analysis, CEA showed to be a significant risk factor of breast cancer, while there was insufficient evidence for an association between HRV parameters (indexed as Awake LF) and breast cancer (odds ratio (O.R.) and 95% confidence interval (95%CI): O.R. = 3.298, 95%CI: 1.156–9.413, p = 0.026) (Table 2).

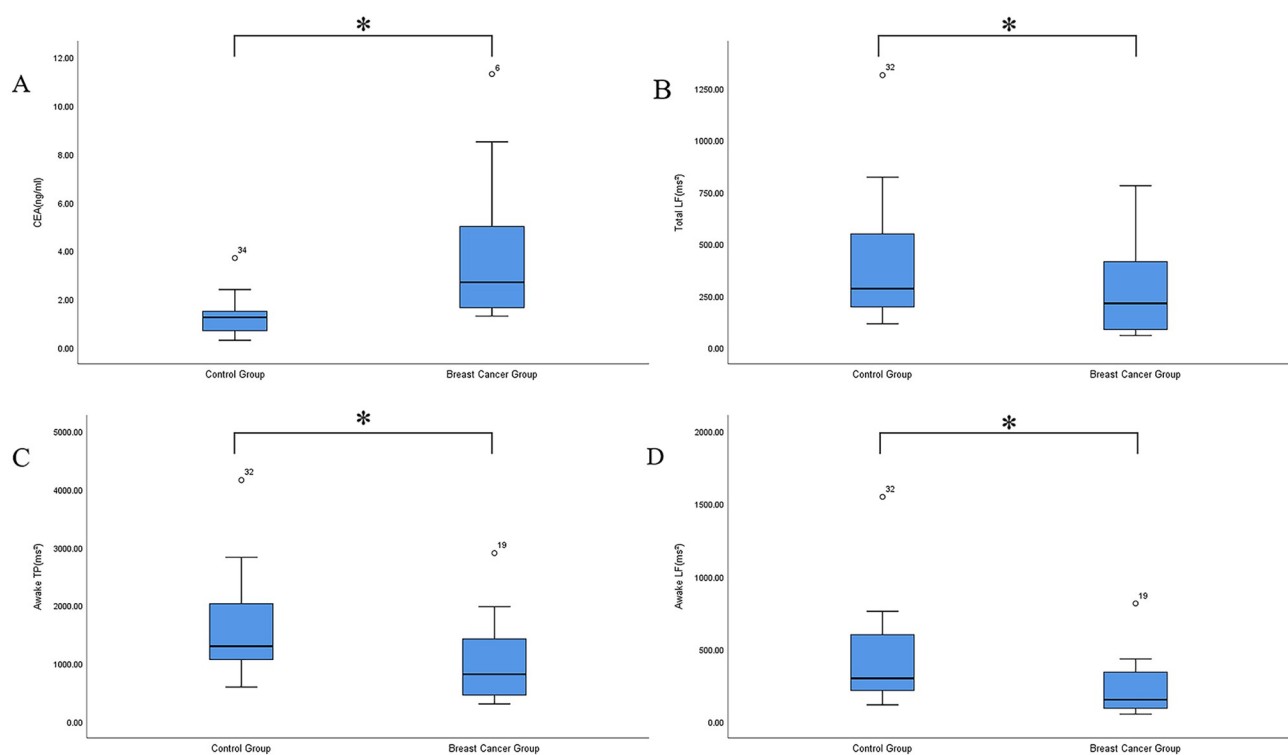

**Fig 2. Mean serum CEA, total LF, awake TP, and awake LF of the groups.**

## Correlation analysis of HRV parameters and CEA

Spearman's correlation analysis showed that total LF, awake TP, and awake LF were negatively correlated with the CEA index in both groups (P<0.05). Given that CEA has good diagnostic performance for breast cancer as a risk factor [29] and differences in HRV parameters between groups, CEA and HRV parameters may have a joint diagnostic effect on breast cancer (S5 Table and Fig 3).

## Analysis of the diagnostic efficacy of serum CEA and HRV parameters in breast cancer

ROC curve analysis showed that only AUC of awake TP, awake LF, and CEA are greater than 0.7. It can be used as a predictor of breast cancer, but the diagnostic efficacy is not high. We combined awake TP, awake LF, and CEA separately as a new combined diagnostic model, and

**Table 2. Logistic regression analysis of factors independently associated with breast cancer.**

| Variables | B-value | P-value | SE | OR | 95%CI |
|---|---|---|---|---|---|
| CEA (ng/ml) | 1.193 | 0.026 | 0.535 | 3.298 | 1.156–9.413 |
| Awake LF (ms$^2$) | -0.002 | 0.377 | 0.002 | 0.998 | 0.994–1.002 |

The P-value of Hosmer-Lemeshow Test is 0.231 (P>0.05), representing the fitting is good (S3 Table).

The colinearity diagnosis found that there was a strong colinearity between HRV parameters, so only CEA and conscious LF were selected for binary logistic analysis (S4 Table).

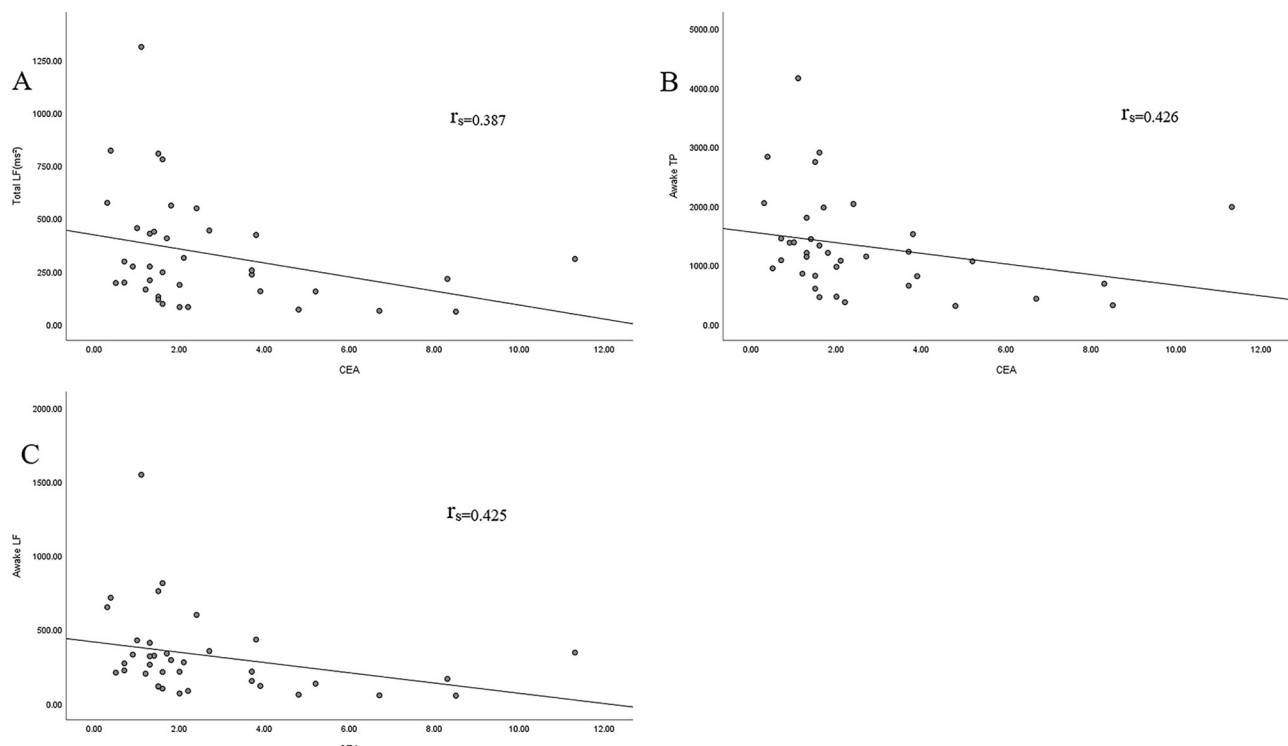

**Fig 3. The association between CEA and total LF, awake TP, awake LF.** (A-C) Scatterplot of correlation analysis of CEA with (A)total LF (P = 0.018); (B)awake LF (P = 0.009); (C)awake TP (P = 0.009).

then incorporated logistic regression models to derive predictive probabilities, and finally performed ROC curve analysis. The results showed that the combination of awake TP, awake LF and CEA had the largest AUC. Its AUC is 0.901 (p<0.001) with 73.7% sensitivity and 94.4% specificity, suggesting a high diagnostic value for breast cancer (Table 3 and Fig 4).

## Discussion

Breast cancer occurred when breast epithelial cells undergo uncontrolled proliferation in response to multiple oncogenic factors. It is widely believed that the incidence of breast cancer has been increasing in recent years. Combining our results with earlier findings, we confirmed the close association of HRV and CEA with tumor development. Our results also showed that the combined analysis of heart rate variability and serum CEA may have clinical value in the early diagnosis and treatment of breast cancer.

**Table 3. Diagnostic efficacy of HRV parameters, CEA, and combined prediction for breast cancer.**

| Variables | Cut-off value | Sensitivity (%) | Specificity (%) | P-value | 95%CI |
|---|---|---|---|---|---|
| Total LF (ms$^2$) | 158.85 | 0.944 | 0.474 | 0.045 | 0.522–0.864 |
| Awake TP (ms$^2$) | 833.50 | 0.944 | 0.526 | 0.033 | 0.534–0.876 |
| Awake LF (ms$^2$) | 183.30 | 0.944 | 0.579 | 0.019 | 0.556–0.895 |
| CEA (ng/ml) | 1.55 | 0.895 | 0.778 | 0.000 | 0.776–0.990 |
| Joint prediction[a] | - | 0.737 | 0.944 | 0.000 | 0.803–0.998 |

[a] Joint prediction refers to combined awake TP, awake LF and CEA (S6 Table).

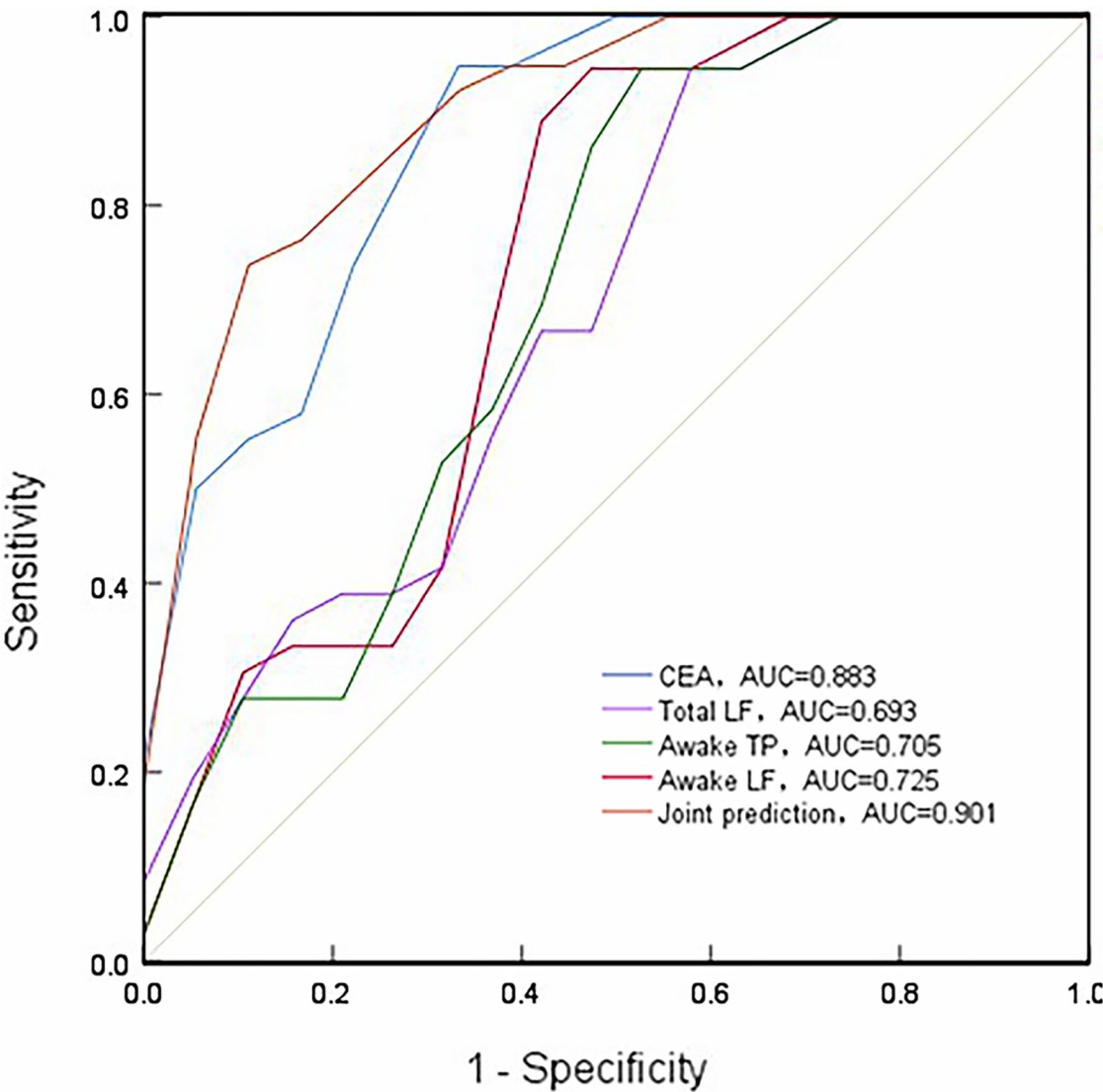

**Fig 4. Receiver operating characteristic (ROC) curves for HRV parameters and CEA.** Joint prediction, combined awake TP, awake LF and CEA; AUC, area under the ROC curve.

Heart rate variability (HRV) is the difference between consecutive R-R intervals during normal heartbeats which also acts as a noninvasive index used to assess the activity of the autonomic nervous system. Studies have shown that HRV had been considered to be closely related to the assessment of disease, risk stratification, treatment outcome, and long-term prognosis of various cardiovascular diseases [30–32]. In recent years, an increasing number of researchers had also applied HRV in oncology. They considered it to be used as a reliable indicator to evaluate the prognosis of cancer patients [33]. And the reduction of heart rate variability in cancer patients was related to shorter survival time [34]. Therefore, our study focused on the relationship between breast cancer and HRV. We selected 19 breast cancer patients and 18 healthy individuals for comparison of 24h ambulatory ECG HRV indicators. The HRV indicators of total LF, awake TP, and awake LF were significantly lower in the breast cancer group than in the control group (P<0.05), where TP represents the level of autonomic nervous system

activity and LF reflects the joint action of sympathetic and some parasympathetic nerves [35]. The results of this study suggested the presence of autonomic dysfunction in breast cancer patients, and similar results have been reported in other studies. Wu [36] showed that HRV was significantly lower in patients with advanced breast cancer and early breast cancer compared with patients with benign breast tumors. And there was a correlation between HRV and TNM stage of breast cancer. Patients with advanced breast cancer have lower HRV, autonomic dysfunction, and possibly poorer prognosis which embodied the help of HRV to construct an effective early diagnosis and clinical prognosis model for breast cancer. Liang [37] applied HRV to the assessment of changes in cardiac autonomic function in breast cancer patients treated with postoperative chemotherapy, reflecting that HRV had a certain guiding significance for the evaluation of cardiac damage in patients. Daniel et al [38] found significant changes in HRV of breast cancer patients and lower parasympathetic cardiac activity compared to the controls, which may be related to the fact that the autonomic nervous system played an important role in the development and progression of cancer [39]. In conclusion, the analysis of heart rate variability in breast cancer patients suggested the presence of autonomic dysfunction and showed significant changes in HRV indicators, which speculated from this result that HRV is of great value for the diagnosis of breast cancer.

Carcinoembryonic antigen (CEA) is a kind of cell membrane structural protein specific for human embryonic antigens secreted by mucosal epithelial cells [40, 41]. And it is also one of the most widely used serum tumor markers in the diagnosis and research of malignant tumors. In our study, we found that serum CEA was significantly higher in breast cancer patients than in healthy people, and that serum CEA had a low negative correlation with total LF, total awake power, and awake LF (p <0.05). CEA in the membranes of tumor cells differentiated from endodermal cells and involved in cell adhesion and regulation processes, so high CEA expression level may be related to a high potential for tumor cell migration and metastasis [41]. Compared with the control group, breast cancer patients showed a significant decrease in awake LF, indicating an increased sympathetic tone in breast cancer patients and some degree of autonomic dysfunction in the heart. Previously studies have found the interaction of the nervous system with the tumor microenvironment was a key regulator for cancer genesis and progression. The sympathetic and parasympathetic nerves in the tumor microenvironment usually regulated cancer development or metastasis through a neurotransmitter-dependent signaling cascade [42]. Additionally, peripheral nervous system activity as a stress response to low HRV indicators caused an elevated expression of noradrenaline levels in breast cancer patients [23]. And behavioral or physiological stressors could promote tumor growth and metastasis by activating tumor β-AR [43]. Sympathetic nerve endings could secrete or locally release stress hormones in the tumor microenvironment, which may directly affect tumor cells and promote their malignant properties. Specifically, norepinephrine and epinephrine could promote tumor cell proliferation, survival (anti-apoptosis), migration, invasion, epithelial-mesenchymal transition (EMT), and production of prostaglandins and matrix metalloproteinases (MMPs) in vitro. Breast cancer cells transferred to the brain received neurotransmitter activity-dependent neurotransmitter signals that triggered a receptor-mediated signaling cascade to induce inward currents in malignant cells, thus driving the development of breast cancer brain metastases [36, 43]. Moreover, the neurological impact of cancer was bidirectional as cancer may induce neurological remodeling and dysfunction. Tumors can secrete neuronal growth factors that increase sympathetic innervation of the tumor. So this created a feedforward cycle in which elevated tumor local noradrenaline levels under a stress-induced sympathetic activation state can promote cancer progression [43]. And since HRV was produced via the sympathetic and parasympathetic-related actions of the autonomic nervous system. Therefore, cancer may affect patient HRV by modulating sympathetic nerves. In summary, the

nervous system may regulate the development or metastasis of cancer, while cancer may induce remodeling and dysfunction of the nervous system. Thus, as breast cancer progresses and cancer cells migrate and metastasize, patients have decreased HRV and increased levels of serum CEA, which is widely present in tumor cell membranes. However, in this study, the low negative correlation of awake LF may be related to the small sample size without excluding mental state, respiratory rate, drug use, and environmental factors.

In this study, binary logistic regression analysis showed that every 1 Hz decrease in awake LF, a protection factors for breast cancer, increased the risk of breast cancer by 2.0%;every 1 μg/L increase in serum CEA, a risk factors for breast cancer, increased the risk of breast cancer by 289.0%. However, total awake power, awake LF, and serum CEA alone only had certain accuracy in the diagnosis of breast cancer, while total LF had low accuracy and weak specificity in the diagnosis of breast cancer in ROC curve analysis. Earlier findings have shown a correlation between increased serum CEA levels and decreased [25, 44] HRV and tumor malignancy. The degree of tumor malignancy was directly proportional to the CEA level, while the CEA level in the early stage of cancer was not obvious [44, 45]; the reduction of HRV was more obvious in later stages of the disease [46]. In other words, separate analysis of serum CEA and HRV was not significant for the early diagnosis of breast cancer. Therefore, to assess this topic, we considered whether the combination of HRV and serum CEA can assist the clinical diagnosis of breast cancer at an early stage and improve its detection rate and the results represented that the diagnostic AUC of combining awake TP, awake LF, and serum CEA were 0.901 and the specificity was 0.944. For its high accuracy and specificity, we inferred that the combination of HRV and serum CEA can assist in the clinical diagnosis of breast cancer at an early stage and improve its detection rate.

## Conclusions

In conclusion, HRV in breast cancer patients suggested abnormal autonomic function. Our study was prospective and we found that total LF, awake TP, and awake LF were negatively correlated with the CEA index (P < 0.05). And serum CEA was a risk factor for breast cancer. In addition, the combination of awake TP, awake LF, and serum CEA had high accuracy in the diagnosis of breast cancer. The combined analysis of heart rate variability and serum CEA may have a predictive effect on the development of breast cancer, and thus should be considered as important indicators for clinical diagnosis and treatment.

## Limitations

In this study, we investigated the changes of HRV and carcinoembryonic antigen in breast cancer patients to provide new ideas for their diagnosis in breast cancer. By analyzing, we came to a clearer conclusion that the combination of HRV and serum CEA can assist in the clinical diagnosis of breast cancer at an early stage and improve its early detection rate, thus implementing early intervention and early treatment and reducing the chance of the disease developing to the middle and late stages. There is little research in this area.

However, there were still several limitations of this study. First, the subjects of this study were obtained from the data of 37 patients in Zhujiang Hospital, and the findings may only be applicable to a small sample size of Asian population which may not be sufficient to detect significant associations between cancer and these HRV indicators, leading to biased conclusions. So the expanded sample size to conduct a more reasonable study needed to be explored in future studies.

Arab et al [47] showed that patients with advanced breast cancer had lower levels of parasympathetic regulation, an imbalance of autonomic nerves, and may be at increased risk for cardiovascular disease compared to patients with early breast cancer. The stage of breast cancer

can affect HRV levels to some extent, whereas the present study did not group the study population by breast cancer stage.

Another limitation of the study was that some breast cancer patients have undergone surgery and chemotherapy. However, factors including preoperative depression, anxiety [46] and cardiotoxicity caused by chemotherapeutic drugs [48] may affect HRV to varying degrees. These influences could not be excluded in this study due to the small sample size. Therefore, a large number of samples from clinical trials are still needed to reduce the interference of other factors with HRV.

## Supporting information

**S1 Data.**
(XLSX)

**S1 Table. Results of normality test.** In the grouping column, "1" represents the breast cancer group and "0" represents the control group.
(PDF)

**S2 Table. Comparison of HRV and CEA indicators between the two groups.** [a] All groups of this index followed a normal distribution and were expressed as $\bar{x} \pm s$. Independent samples t-test was used for comparison between groups. [b] At least one of the groups of this indicator did not follow a normal distribution, denoted by M (P25, P75), and comparisons between groups were made using rank sum test.
(PDF)

**S3 Table. The results of Hosmer-Lemeshow test.**
(PDF)

**S4 Table. Results of collinearity regression.** SE, standard error; D-W value, the indicators of Durbin-Watson test. -7.550E-5 represents $-7.550 \times 10^{-5}$.
(PDF)

**S5 Table. Correlation between HRV and CEA.**
(PDF)

**S6 Table. Joint prediction of awake TP, awake LF and CEA.**
(PDF)

## Acknowledgments

We would like to thank the Zhujiang Hospital of Southern Medical University, Guangdong, China for providing the data.

## Author Contributions

**Conceptualization:** Lishan Ding, Mingsi Chi, Yaping Huang, Lei He.

**Data curation:** Lishan Ding, Yuepeng Yang, Mingsi Chi, Zijun Chen, Yaping Huang, Wenshan Ouyang.

**Formal analysis:** Lishan Ding, Yuepeng Yang, Mingsi Chi, Yaping Huang, Wenshan Ouyang.

**Methodology:** Lishan Ding, Yuepeng Yang, Zijun Chen, Lei He.

**Resources:** Lei He, Ting Wei.

**Software:** Lishan Ding, Yuepeng Yang, Zijun Chen.

**Supervision:** Lei He, Ting Wei.

**Validation:** Lishan Ding, Ting Wei.

**Visualization:** Lishan Ding, Zijun Chen, Lei He.

**Writing – original draft:** Lishan Ding, Yuepeng Yang, Mingsi Chi, Zijun Chen, Yaping Huang, Wenshan Ouyang.

**Writing – review & editing:** Lishan Ding, Weijian Li, Lei He, Ting Wei.

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
