## [Decision Letter · Decision Letter 0]

20 Dec 2022

PONE-D-22-29053Diagnostic role of heart rate variability in breast cancer and its relationship with peripheral serum carcinoembryonic antigenPLOS ONE

Dear Dr. He,

Thank you for submitting your manuscript to PLOS ONE. After careful consideration, we feel that it has merit but does not fully meet PLOS ONE’s publication criteria as it currently stands. Therefore, we invite you to submit a revised version of the manuscript that addresses the points raised during the review process.

We look forward to receiving your revised manuscript.

Kind regards,

Alessandro Rizzo

Academic Editor

PLOS ONE

Journal Requirements:

2. Our staff editors have determined that your manuscript is likely within the scope of our Early Detection, Screening and Diagnosis of Cancer Call for Papers. This editorial initiative is headed by in-house PLOS editors. This Call for Papers aims to explore recent advances in the early detection of cancer and implications of these advances for patient survival. Additional information can be found on our announcement page: https://collections.plos.org/call-for-papers/early-detection-screening-and-diagnosis-of-cancer/

If you would like your manuscript to be considered for this collection, please let us know in your cover letter and we will ensure that your paper is treated as if you were responding to this call. Please note that being considered for the Call for Papers does not require additional peer review beyond the journal’s standard process and will not delay the publication of your manuscript if it is accepted by PLOS ONE. If you would prefer to remove your manuscript from collection consideration, please specify this in the cover letter.

Reviewers' comments:

Reviewer's Responses to Questions

**Comments to the Author**

1. Is the manuscript technically sound, and do the data support the conclusions?

Reviewer #1: Partly

Reviewer #2: Partly

Reviewer #3: No

Reviewer #4: Partly

2. Has the statistical analysis been performed appropriately and rigorously? 

Reviewer #1: Yes

Reviewer #2: Yes

Reviewer #3: No

Reviewer #4: Yes

3. Have the authors made all data underlying the findings in their manuscript fully available?

Reviewer #1: Yes

Reviewer #2: Yes

Reviewer #3: No

Reviewer #4: Yes

4. Is the manuscript presented in an intelligible fashion and written in standard English?

Reviewer #1: No

Reviewer #2: No

Reviewer #3: Yes

Reviewer #4: No

5. Review Comments to the Author

Reviewer #1: The study assesses a current, timely topic in breast cancer.

We recommend some changes:

- We believe this article is suitable for publication in the journal although major revisions are needed. The main strengths of this paper are that it addresses an interesting and very timely question and provides a clear answer, with some limitations. Certainly, the study is limited to an Asian population with a very small sample size, and authors should further express this point.

- Second, the study included a widely varied patient population from a chinese institute and the total number of patients analyzed was relatively small. Thus, the authors should better highlight the limitations of the current paper.

- The background of the changing scenario of medical treatment in breast cancer patients should be better discussed, and some recent papers regarding this topic should be included in the introduction section (PMID: 34802383; PMID: 36368251 ; PMID: 34793275), only for a matter of consistency. In fact, the introduction appears a but poor and more paragraphs and data are needed to introduce this topic.

Major changes are necessary.

Reviewer #2: Experimental and clinical studies have shown that the sympathetic nervous system (SNS) stimulates cancer progression and reduces the efficacy of oncological treatment. For the determination of SNS modulation, the non-invasive method of heart rate variability (HRV) is widely used. Research articles have been published addressing the clinical value of HRV in breast cancer patients since 1999. However, the small sample size and heterogeneity, the presence of confounders, and the observational study design are the limitations of those studies. The main findings from these studies included the prognostic value of HRV detection and it is revealing the SNS modulation in breast cancer survivors. Few studies have been reported about the value of HRV detection in the early diagnosis of breast cancer. Here, this study demonstrated that the combined HRV and serum CEA analysis might have a predictive effect on the development of breast cancer. However, a major revision of the study design and sample size should be awared before being accepted.

Major Issues

1. Although the idea of this research has novelty for the combination of HRV and CEA in early diagnosis of breast cancer, the sample size is too small which is not compelling for the conclusion. In that case, I would like to suggest you provide more solid data.

2. HRV could be influenced by several important factors in breast cancer survivors, eg. cardiotoxicity of chemotherapy and/or radiation therapy, surgery-induced fatigue, and stress. I suggest the author provide that information on enrolled breast cancer patients and discuss the potential influence on the results.

3. If you are trying to convince the aberrant HRV could be a new potential biomarker for diagnosis of breast cancer, the control group should include age-matched women with benign breast disease.

Minor Issues

1. Since HRV has been reported to be associated with the stage of cancer, they'd better provide information on breast cancer staging and timing of collecting the blood sample and measuring the HRV as well.

2. Please consider describing the limitations of your research in the discussion section. For example, a small sample size may cause bias in the conclusion but you are going to enroll more participants to study in the future.

3. Please simplify or precise the notes under all figures.

4. In the introduction section, please add more experimental and clinical evidence of HRV in breast cancer. For example, consider citing some associated literature.

1) Arab C, Dias DP, Barbosa RT, Carvalho TD, Valenti VE, Crocetta TB, Ferreira M, Abreu LC, Ferreira C. Heart rate variability measure in breast cancer patients and survivors: A systematic review. Psychoneuroendocrinology. 2016 Jun;68:57-68. doi: 10.1016/j.psyneuen.2016.02.018.

2) Majerova K, Zvarik M, Ricon-Becker I, Hanalis-Miller T, Mikolaskova I, Bella V, Mravec B, Hunakova L. Increased sympathetic modulation in breast cancer survivors determined by measurement of heart rate variability. Sci Rep. 2022 Aug 29;12(1):14666. doi: 10.1038/s41598-022-18865-7.

5. Please explain the reason why you did not choose other serum tumor markers, like CA153 instead of CEA.

6. Several sentences are tediously written and could be shortened.

7. I may suggest authors

8. The format of references should be corrected.

Reviewer #3: Sample size is not big enough in this study, to ensure the sample is sufficiently representative, the number of samples selected is modified to the needs of the statistical analysis. According to cross-sectional study, for age part, as SD set 11.82, the sample size for each group should be 42, as SD set 12.76, the sample size for each group should be 48.　Comparative studies, such as comparing experimental and control groups: 30 samples are required at least for each group (Gay, 1992).

For the missing data, it is over 20 percent. The control group is selected from 562 to 18, the breast cancer group is selected from 229 to 19 in the final. If too much original data is missing, not only will the statistical power be reduced, the standard error will become larger, and even the information of the data will be distorted or misleading; it will also make the correlation coefficient matrix or covariate matrix Estimates are biased, which leads to biases in the extraction of common factors, the resulting factors are different from the actual situation.

Reviewer #4: In "Diagnostic role of heart rate variability in breast cancer and its relationship with peripheral serum carcinoembryonic antigen" Ding et al. investigated the role of heart rate variability (HRV) and serum carcinoembryonic antigen (CEA) in breast cancer prediction. Early diagnosis of breast cancer is critical and it's an interesting study to identify potential new biomarkers to identify breast cancer patients. However, there are a few major issues to be addressed before the manuscript is published in high impact journals.

Major:

1. I recommend the authors get editing help to improve the academic writing for this manuscript. I can appreciate the findings as it is, but I believe the manuscript can benefit from professional writing to reach its full potential.

2. The introduction lacks the literature and key references. A comprehensive introduction is essential for a paper that it bridges the gap between your readers and your own research. In this part, the manuscript is expected to guide the readers the key milestones in HRV, CEA and prediction of breast cancer, and also the rationale behind your hypothesis (the use of HRV and CEA can predict breast cancer at early stage.) The authors briefly introduced the background for HRV and CEA, however, their links to breast cancer are not clear.

3. Novelty of this work. To publish as an original article, the novelty of this work needs to be emphasized so the readers can fully appreciate the importance of your work.

4. The results section is not well organized. The authors need to state clearly the purpose of each experiment, the observations and conclusions. For example, line 174, the authors examined the correlation between HRV and CEA. The authors have already identified differences of HRV and CEA between patient to control groups. What questions do the authors would like to address by examining the relationship between HRV and CEA and what does the negative correlation between HRV and CEA mean in the context of breast cancer?

5. Joint prediction using combined markers have shown to improve the prediction performance. What computational models did the authors use to the markers (awake TP, awake LF and CEA) are not shown.

6. How to validate the findings in this manuscript? Have the authors consider independent studies as validation datasets?

6. PLOS authors have the option to publish the peer review history of their article (what does this mean?). If published, this will include your full peer review and any attached files.

Reviewer #1: No

Reviewer #2: No

Reviewer #3: No

Reviewer #4: No

---

## [Author Response · Author response to Decision Letter 0]

29 Jan 2023

Dear Editors and Reviewers:

Thank you for your letter and for the reviewers’ comments concerning our manuscript entitled “Diagnostic role of heart rate variability in breast cancer and its relationship with peripheral serum carcinoembryonic antigen” (ID: PONE-D-22-29053). Those comments are all valuable and very helpful for revising and improving our paper, as well as the important guiding significance to our research. We have studied comments carefully and have made the correction which we hope meet with approval. The revised portion is marked in the paper. The main corrections in the paper and the responses to the reviewer’s comments are as flowing:

Response to Journal Requirements: Thank you for your positive comments on our manuscript. We have responded to each of your points in the followings.

1.Please ensure that your manuscript meets PLOS ONE's style requirements, including those for file naming.

Response1：We are very sorry for our incorrect writing. We have modified the style requirements of the manuscript according to the requirements of the PLOS ONE style template so that it meets the publication requirements of your journal.

2.Our staff editors have determined that your manuscript is likely within the scope of our Early Detection, Screening and Diagnosis of Cancer Call for Papers. This editorial initiative is headed by in-house PLOS editors. This Call for Papers aims to explore recent advances in the early detection of cancer and implications of these advances for patient survival. If you would like your manuscript to be considered for this collection, please let us know in your cover letter and we will ensure that your paper is treated as if you were responding to this call. 

Response2：Thank the reviewer for the constructive comments and suggestions. We consider our manuscript is suitable for the scope of your Early Detection, Screening and Diagnosis of Cancer Call for Papers. We agree to include the manuscript in this collection.

Response to Reviewer 1: Thank you for your review of our paper. We have answered each of your points below.

1.The study assesses a current, timely topic in breast cancer. We believe this article is suitable for publication in the journal although major revisions are needed. The main strengths of this paper are that it addresses an interesting and very timely question and provides a clear answer, with some limitations.

Response 1：We thank the reviewer for his/her positive comments on our paper. As Reviewer suggested, we have noted those individually in the specific cases below.

2. The study is limited to an Asian population with a very small sample size, and authors should further express this point. The study included a widely varied patient population from a Chinese institute and the total number of patients analyzed was relatively small. Thus, the authors should better highlight the limitations of the current paper.

Response 2: Thanks for your kind suggestions, which is valuable for improving the accuracy of the manuscript. It is really true as Reviewer suggested that the study is limited to an Asian population with a very small sample size. So we have made modifications and further emphasized this limitation in the manuscript.

However, there were still several limitations of this study. First, the subjects of this study were obtained from the data of 37 patients in Zhujiang Hospital, and the findings may only be applicable to a small sample size of Asian population which may not be sufficient to detect significant associations between cancer and these HRV indicators, leading to biased conclusions. So the expanded sample size to conduct a more reasonable study needed to be explored in future studies. (p. 18, lines 238-332)

Because our hospital is located in Asia, the population of our study is primarily Asian.

As mentioned, our study population was drawn from a widely varied patient population from the hospital. In order to make the results reliable, we ensured the homogeneity of the enrolled cases through strict inclusion criteria and exclusion criteria as follows.

Inclusion criteria: 1) patients with complete general clinical information; 2) meeting the diagnostic criteria for breast cancer lesions. 

Exclusion criteria: 1) organic heart disease such as heart failure; 2) pre-existing palpitations, abnormal heart rate, etc.; 3) hyperthyroidism; 4) diabetes mellitus; 5) inflammation; 6) infection.(p. 5-6, lines 103-106)

In addition, the number of patients with breast cancer who have a clinical examination such as a 24-hour ambulatory ECG is quite small, which makes it difficult to obtain a larger sample size by reviewing medical records.

However, it does not mean that our present study is unreliable. Like some literature, Karolina Majerova et al.[1] studied sympathetic nerve activity in breast cancer survivors by measurement of HRV. Their study sample consisted of four groups. The healthy women control group consisted of 21 randomly selected women. The group of patients with benign tumors consisted of 13 women. The group of patients with active breast cancer consisted of 20 women with newly diagnosed breast cancer, and the group of survivors consisted of 15 women. Despite their small number of subjects per group, their study was recognized by the journal through rigorous screening and inclusion criteria (IF:4.996).

In addition, we calculated the sample sizes with an actual power of 0.91153using the PASS (v. 15), a powerful sample size estimation software[2]. Notably, the power is the probability of rejecting a false null hypothesis. The parameters are shown in Response Fig 1A. The result showed that the sample size was 8 in each group (Response Fig 1B). Notably, patients with breast cancer who also had a 24-hour ambulatory ECG are indeed difficult to find clinically, so we are currently unable to expand the sample size further by reviewing medical records. However, the current sample size exceeds the number estimated by the PASS software.

Response Fig 1. The result of sample size calculation. (A) The parameters. (B) The result of calculation.

In summary, the small sample size of our study is indeed a limitation, but we believe that our strict inclusion and exclusion criteria will make our study comparable and reliable. In the future, we will continue to accumulate relevant clinical cases and make a higher quality and more reliable study based on this article.

3.The background of the changing scenario of medical treatment in breast cancer patients should be better discussed, and some recent papers regarding this topic should be included in the introduction section (PMID: 34802383; PMID: 36368251; PMID: 34793275), only for a matter of consistency. In fact, the introduction appears a but poor and more paragraphs and data are needed to introduce this topic.

Response 3: Thank you very much for your precious and constructive guidance on our foreword. We have rewritten the introduction and added background on the changing medical situation of breast cancer patients, citing the literature you provided. (p. 3-5, lines 44-88)

First, we reviewed the global cancer update provided by GLOBOCAN 2020 to add the latest incidence and mortality rates for breast cancer. We have added a discussion of the changing status of the current medical situation of breast cancer patients. In summary, breast cancer has become one of the most prevalent malignancies in women in terms of incidence and mortality. Patients with early stage breast cancer are asymptomatic. However, they are often in the middle to late stages when symptoms become apparent, making treatment difficult and prognosis poor. Therefore, we believe that it is important to explore the means of early diagnosis of breast cancer.

Secondly, we add the reasons why CEA has been chosen as an indicator for the early diagnosis of breast cancer and cite studies by others to support our claims. Firstly, among the methods of early diagnosis of breast cancer, CEA can better reflect the development of the tumour itself. Second, CEA, as a tumour marker, can be obtained by simply collecting venous blood, which has the characteristics of being rapid and easy to collect. Thirdly, monitoring CEA is basically harmless to human body and at the same time highly sensitive. Many studies in recent years have shown that preoperative CEA levels may provide useful information for the identification and treatment of breast cancer. The European Tumour Markers Panel recommends CEA levels as an indicator for the assessment of prognosis, early detection of disease progression and monitoring of treatment in breast cancer patients. We therefore believe that the use of CEA for the early diagnosis of breast cancer is possible. However, as the specificity of CEA for the early diagnosis of breast cancer is low, it is necessary to combine it with other diagnostic methods to improve the diagnostic efficacy.

Thus, combined with clinical experience, we found significant differences between HRV in breast cancer patients and controls, and by referring to the literature we found that HRV is widely used in clinical practice as a non-invasive measure to assess autonomic nervous system activity. The results of Karolina Majerova et al. showed that sympathetic regulation was significantly increased in breast cancer patients compared to healthy volunteers. The results suggest that sympathetic inhibition can inhibit the development of breast cancer. In summary, we conclude that HRV may be a clinical tool for detecting early breast cancer.

Therefore, the aim of this study was to investigate heart rate variability and carcinoembryonic antigen changes in breast cancer patients and their role in the diagnosis of breast cancer, to provide a new complementary method for the early diagnosis of breast cancer and to improve the detection rate.

Response to Reviewer 2: Thank you for your comments. Our answers to your points are as follows.

Major Issues

1.Although the idea of this research has novelty for the combination of HRV and CEA in early diagnosis of breast cancer, the sample size is too small which is not compelling for the conclusion. In that case, I would like to suggest you provide more solid data.

Response 1: Thank you very much indeed for your comments and your confirmation of the novelty of our manuscript. 

As you mentioned, the small sample size is the limitation of our manuscript. I am sorry that the number of patients with breast cancer who have a clinical examination such as a 24-hour ambulatory ECG is quite small, which makes it difficult to obtain a larger sample size by reviewing medical records.

We have made modifications and further emphasized this limitation in the manuscript.

However, there were still several limitations of this study. First, the subjects of this study were obtained from the data of 37 patients in Zhujiang Hospital, and the findings may only be applicable to a small sample size of Asian population which may not be sufficient to detect significant associations between cancer and these HRV indicators, leading to biased conclusions. So the expanded sample size to conduct a more reasonable study needed to be explored in future studies. (p. 18, lines 238-332)

However, it does not mean that our present study is unreliable. Like some literature, Karolina Majerova et al.[1] studied sympathetic nerve activity in breast cancer survivors by measurement of HRV. Their study sample consisted of four groups. The healthy women control group consisted of 21 randomly selected women. The group of patients with benign tumors consisted of 13 women. The group of patients with active breast cancer consisted of 20 women with newly diagnosed breast cancer, and the group of survivors consisted of 15 women. Despite their small number of subjects per group, their study was recognized by the journal through rigorous screening and inclusion criteria (IF:4.996).

In addition, we calculated the sample sizes with an actual power of 0.91153using the PASS (v. 15), a powerful sample size estimation software[2]. Notably, the power is the probability of rejecting a false null hypothesis. The parameters are shown in Response Fig 1A. The result showed that the sample size was 8 in each group (Response Fig 1B). Notably, patients with breast cancer who also had a 24-hour ambulatory ECG are indeed difficult to find clinically, so we are currently unable to expand the sample size further by reviewing medical records. However, the current sample size exceeds the number estimated by the PASS software.

In summary, the small sample size of our study is indeed a limitation, but we believe that our strict inclusion and exclusion criteria will make our study comparable and reliable. In the future, we will continue to accumulate relevant clinical cases and make a higher quality and more reliable study based on this article.

2.HRV could be influenced by several important factors in breast cancer survivors, eg. cardiotoxicity of chemotherapy and/or radiation therapy, surgery-induced fatigue, and stress. I suggest the author provide that information on enrolled breast cancer patients and discuss the potential influence on the results

Response2：Thank you very much indeed for your comments. We understand that the interference factors in HRV might exist as the reviewer stated. Information on radiotherapy, chemotherapy, surgery and staging of breast cancer patients is provided in the excel 'data'. In addition, we added a statement in our revised manuscript to reflect this point of the reviewer and will pay more attention on this question in our future studies.

Arab et al showed that patients with advanced breast cancer had lower levels of parasympathetic regulation, an imbalance of autonomic nerves, and may be at increased risk for cardiovascular disease compared to patients with early breast cancer. The stage of breast cancer can affect HRV levels to some extent, whereas the present study did not group the study population by breast cancer stage.

Another limitation of the study was that some breast cancer patients have undergone surgery and chemotherapy. However, factors including preoperative depression, anxiety and cardiotoxicity caused by chemotherapeutic drugs may affect HRV to varying degrees. These influences could not be excluded in this study due to the small sample size. Therefore, a large number of samples from clinical trials are still needed to reduce the interference of other factors with HRV. (p.18, lines 333-341).

3.If you are trying to convince the aberrant HRV could be a new potential biomarker for diagnosis of breast cancer, the control group should include age-matched women with benign breast disease.

Response3: We thank for the reviewer for pointing out this issue. We agree with this suggestion and have searched the articles on HRV and autonomic nervous system in patients with benign breast disease, but we disappointedly found that the relationship between them was not clear. In fact, this is an exciting future area of investigation for us. It is known that the later the stage of malignant tumor patients, the more obvious the reduction of heart rate variability[3]. This may suggest that patients with benign breast disease have less variation in heart rate. In addition, in our hospital, patients with benign breast disease received fewer samples for HRV analysis, which may increase the uncertainty of the results. 

Minor Issues

1.Since HRV has been reported to be associated with the stage of cancer, they'd better provide information on breast cancer staging and timing of collecting the blood sample and measuring the HRV as well.

Response1: Thank you very much indeed for your comments. We had provided information on breast cancer staging and timing of collecting the blood sample and measuring the HRV in the excel 'data'. Of the subjects enrolled in the study, we selected the timing of the collection of blood samples and the measurement of HRV to be on essentially the same day. We have added as much information as possible on the staging of breast cancer. However, due to the small number of cases at each stage, we are unable to carry out a study on the relationship between CEA and HRV in patients with different stages of breast cancer at this time.

2.Please consider describing the limitations of your research in the discussion section. For example, a small sample size may cause bias in the conclusion but you are going to enroll more participants to study in the future

Response2: We greatly appreciate your valuable suggestions for the discussion section of our article. We understand that bias of a small sample size might exist as the reviewer stated. In the future, we will continue to accumulate relevant cases or recruit more participants for the study in order to improve the credibility and accuracy of the article. We have made modifications and further emphasized the limitation in the manuscript.

However, there were still several limitations of this study. First, the subjects of this study were obtained from the data of 37 patients in Zhujiang Hospital, and the findings may only be applicable to a small sample size of Asian population which may not be sufficient to detect significant associations between cancer and these HRV indicators, leading to biased conclusions. So the expanded sample size to conduct a more reasonable study needed to be explored in future studies.

Arab et al showed that patients with advanced breast cancer had lower levels of parasympathetic regulation, an imbalance of autonomic nerves, and may be at increased risk for cardiovascular disease compared to patients with early breast cancer. The stage of breast cancer can affect HRV levels to some extent, whereas the present study did not group the study population by breast cancer stage.

Another limitation of the study was that some breast cancer patients have undergone surgery and chemotherapy. However, factors including preoperative depression, anxiety and cardiotoxicity caused by chemotherapeutic drugs may affect HRV to varying degrees. These influences could not be excluded in this study due to the small sample size. Therefore, a large number of samples from clinical trials are still needed to reduce the interference of other factors with HRV. (p.18, lines 328-341).

3.Please simplify or precise the notes under all figures.

Response3: Thank you very much for your guidance on the notes and details of our articles. We have precise the notes under all figures. We have simplified the notes under all the figures to make the article more concise and precise.

4.In the introduction section, please add more experimental and clinical evidence of HRV in breast cancer. For example, consider citing some associated literature. 1) Arab C, Dias DP, Barbosa RT, Carvalho TD, Valenti VE, Crocetta TB, Ferreira M, Abreu LC, Ferreira C. Heart rate variability measure in breast cancer patients and survivors: A systematic review. Psychoneuroendocrinology. 2016 Jun;68:57- 68. doi: 10.1016/j.psyneuen.2016.02.018. 2) Majerova K, Zvarik M, Ricon-Becker I, Hanalis-Miller T, Mikolaskova I, Bella V, Mravec B, Hunakova L. Increased sympathetic modulation in breast cancer survivors determined by measurement of heart rate variability. Sci Rep. 2022 Aug 29;12(1):14666. doi: 10.1038/s41598-022-18865-7.

Response4: Thank you very much for your precious and constructive guidance on our foreword. We have rewritten the introduction and added more experimental and clinical evidence of HRV in breast cancer, citing the literature you provided.

Heart rate variability (HRV) refers to the change between each cardiac cycle, which originates from the autonomic regulation of the heart's sinus node. It is considered to be an important indicator of autonomic function and action, reflecting the balance between the vagus and sympathetic nerves. Studies have found that patients with breast cancer have a higher risk of cardiovascular disease which presented a lower HRV, implying vagal dysfunction. Karolina Majerova et al. showed cardiac vagal modulation alterations in breast cancer survivors by measuring HRV, revealing a significant increase in sympathetic modulation in breast cancer patients relative to healthy volunteers. In addition, previous studies have shown that HRV analysis can help determine tumor staging, efficacy, prognosis, and autonomic function. In Desmond G. Powe's trial, beta-blocker therapy significantly reduced distant metastasis, cancer recurrence, and cancer-specific mortality in breast cancer patients, suggesting that sympathetic inhibition can inhibit breast cancer progression. To be concluded, breast cancer patients have impaired autonomic nervous system activity so early recognition is clinically significant to increase treatment chances and survival time. Therefore, HRV, as a non-invasive measure widely used in clinical practice to assess autonomic nervous system activity, may be a clinical tool for detecting early breast cancer.

In summary, it is insidious and lacks of effective screening methods to diagnose in the early stages of breast cancer, which seriously affects the life and health of women. Therefore, this study aimed to investigate the changes in heart rate variability and carcinoembryonic antigen in breast cancer patients and their role in the diagnosis of breast cancer, to provide a new adjunctive method for early diagnosis of breast cancer, and to improve the detection rate. (p.4-5, lines 70-88).

5. Please explain the reason why you did not choose other serum tumor markers, like CA153 instead of CEA.

Response5：Thank you very much for pointing out this question. CA15-3 and CEA are considered to be the two most valuable tumour markers for the early diagnosis and efficacy of breast cancer due to their high expression and close association with breast carcinogenesis[4]. Wang et al.[5] found that when individual tumour markers were used to diagnose metastatic breast cancer, CEA had the highest sensitivity, relative to CA19-9, CA125, CA15-3 and tissue peptide-specific antigen (TPS). In addition, Li et al.[6] showed that elevated CA15-3 was associated with advanced histological grade and younger age (<35 years), whereas elevated CEA was associated with non-triple-negative tumour types and older age. According to statistics[7], the age of onset of breast cancer is high and the incidence of breast cancer in older women is high, and our study focuses on older patients with early stage breast cancer. In summary, we propose the following hypothesis: CEA may be more appropriate for diagnostic screening of early breast cancer. 

6. Several sentences are tediously written and could be shortened.

Response6：Thank you very much for pointing out this question. We apologize for the long and tedious sentences in the article. Several sentences have been shortened and improved to respond to the reviewer’s comments. As the same time, we have had someone specializing in English to touch up the article. This deficiency has been corrected in the revised manuscript. Please see the revised manuscript.

7.I may suggest authors the format of references should be corrected.

Response7：Thank you very much for your valuable suggestions. We have made modifications to the format of the references.

Response to Reviewer 3: Thank you for your comments. Our answers to your points are as follows.

1.Sample size is not big enough in this study, to ensure the sample is sufficiently representative, the number of samples selected is modified to the needs of the statistical analysis. According to cross-sectional study, for age part, as SD set 11.82, the sample size for each group should be 42, as SD set 12.76, the sample size for each group should be 48. Comparative studies, such as comparing experimental and control groups: 30 samples are required at least for each group (Gay, 1992)

Response1：Thank you very much for your valuable suggestions. For normally distributed data, the Mean and Standard Deviation (SD) are generally used to describe concentrated trends and dispersion, while for non-normally distributed data, the Median and Interquartile Range (IQR) are commonly used to describe concentrated trends and dispersion. Dispersion. In our study, the key indicators of CEA and HRV were statistically analysed to be non-normally distributed and therefore standard deviation is not used to describe them. 

In addition, we calculated the sample sizes with an actual power of 0.91153using the PASS (v. 15), a powerful sample size estimation software[2]. Notably, the power is the probability of rejecting a false null hypothesis. The parameters are shown in Response Fig 1A. The result showed that the sample size was 8 in each group (Response Fig 1B). Notably, patients with breast cancer who also had a 24-hour ambulatory ECG are indeed difficult to find clinically, so we are currently unable to expand the sample size further by reviewing medical records. However, the current sample size exceeds the number estimated by the PASS software.

As you mentioned, the small sample size is the limitation of our manuscript. I am sorry that the number of patients with breast cancer who have a clinical examination such as a 24-hour ambulatory ECG is quite small, which makes it difficult to obtain a larger sample size by reviewing medical records.

We have made modifications and further emphasized this limitation in the manuscript.

However, there were still several limitations of this study. First, the subjects of this study were obtained from the data of 37 patients in Zhujiang Hospital, and the findings may only be applicable to a small sample size of Asian population which may not be sufficient to detect significant associations between cancer and these HRV indicators, leading to biased conclusions. So the expanded sample size to conduct a more reasonable study needed to be explored in future studies. (p. 18, lines 238-332)

However, it does not mean that our present study is unreliable. Like some literature, Karolina Majerova et al. studied sympathetic nerve activity in breast cancer survivors by measurement of HRV. Their study sample consisted of four groups. The healthy women control group consisted of 21 randomly selected women. The group of patients with benign tumors consisted of 13 women. The group of patients with active breast cancer consisted of 20 women with newly diagnosed breast cancer, and the group of survivors consisted of 15 women. Despite their small number of subjects per group, their study was recognized by the journal through rigorous screening and inclusion criteria (IF:4.996).

In summary, the small sample size of our study is indeed a limitation, but we believe that our strict inclusion and exclusion criteria will make our study comparable and reliable. In the future, we will continue to accumulate relevant clinical cases and make a higher quality and more reliable study based on this article.

2.For the missing data, it is over 20 percent. The control group is selected from 562 to 18, the breast cancer group is selected from 229 to 19 in the final. If too much original data is missing, not only will the statistical power be reduced, the standard error will become larger, and even the information of the data will be distorted or misleading; it will also make the correlation coefficient matrix or covariate matrix Estimates are biased, which leads to biases in the extraction of common factors, the resulting factors are different from the actual situation.

Response2: Special thanks to you for your good comments. Data missing value does not exist in the independent variable. Our study only removed samples that did not meet the experimental conditions. In addition, as this index was not included in the examination program in the hospital until 2016, data before 2016 was missing, which may be similar to missing completely at random. Therefore, we adopt the direct elimination method to complete case analysis, which will not bias the evaluation of the results. We will be happy to edit the text further, based on helpful comments from the reviewers.

Response to Reviewer 4: Thank you for your comments. Our answers to your points are as follows.

1. I recommend the authors get editing help to improve the academic writing for this manuscript. I can appreciate the findings as it is, but I believe the manuscript can benefit from professional writing to reach its full potential. 

Response1: Thank you for the constructive comments and suggestions that helped us to greatly improve the manuscript. We have had the article touched up by someone who specialises in English to ensure that it is grammatically correct. This deficiency has been corrected in the revised manuscript. Please see the revised manuscript.

2. The introduction lacks the literature and key references. A comprehensive introduction is essential for a paper that it bridges the gap between your readers and your own research. In this part, the manuscript is expected to guide the readers the key milestones in HRV, CEA and prediction of breast cancer, and also the rationale behind your hypothesis (the use of HRV and CEA can predict breast cancer at early stage.) The authors briefly introduced the background for HRV and CEA, however, their links to breast cancer are not clear.

Response2: Thank you very much for your precious and constructive guidance on our foreword. The introduction has been rewritten to include references to a number of clinical and experimental studies related to breast cancer, and to provide more detail on the link between HRV and CEA and breast cancer. Background information on breast cancer has also been added. (p. 3-5, lines 44-88)

First, we reviewed the global cancer update provided by GLOBOCAN 2020 to add the latest incidence and mortality rates for breast cancer. We have added a discussion of the changing status of the current medical situation of breast cancer patients. In summary, breast cancer has become one of the most prevalent malignancies in women in terms of incidence and mortality. Patients with early stage breast cancer are asymptomatic and by the time symptoms become apparent they are often in the middle to late stages, making treatment difficult and prognosis poor. Therefore, we believe that it is important to explore the means of early diagnosis of breast cancer.

Secondly, we add the reasons why CEA has been chosen as an indicator for the early diagnosis of breast cancer and cite studies by others to support our claims. Firstly, among the methods of early diagnosis of breast cancer, CEA can better reflect the development of the tumour itself. Second, CEA, as a tumour marker, can be obtained by simply collecting venous blood, which has the characteristics of being rapid and easy to collect. Thirdly, monitoring CEA is basically harmless to human body and at the same time highly sensitive. Many studies in recent years have shown that preoperative CEA levels may provide useful information for the identification and treatment of breast cancer. The European Tumour Markers Panel recommends CEA levels as an indicator for the assessment of prognosis, early detection of disease progression and monitoring of treatment in breast cancer patients. We therefore believe that the use of CEA for the early diagnosis of breast cancer is possible. However, as the specificity of CEA for the early diagnosis of breast cancer is low, it is necessary to combine it with other diagnostic methods to improve the diagnostic efficacy.

Thus, combined with clinical experience, we found significant differences between HRV in breast cancer patients and controls, and by referring to the literature we found that HRV is widely used in clinical practice as a non-invasive measure to assess autonomic nervous system activity. The results of Karolina Majerova et al. showed that sympathetic regulation was significantly increased in breast cancer patients compared to healthy volunteers. The results suggest that sympathetic inhibition can inhibit the development of breast cancer. In summary, we conclude that HRV may be a clinical tool for detecting early breast cancer.

Therefore, the aim of this study was to investigate heart rate variability and carcinoembryonic antigen changes in breast cancer patients and their role in the diagnosis of breast cancer, to provide a new complementary method for the early diagnosis of breast cancer and to improve the detection rate.

3. Novelty of this work. To publish as an original article, the novelty of this work needs to be emphasized so the readers can fully appreciate the importance of your work.

Response3: Thank you very much indeed for your comments. We agree with this suggestion, so we have made modifications and further emphasized the novelty of this work in the manuscript.

In this study, we investigated the changes of HRV and carcinoembryonic antigen in breast cancer patients to provide new ideas for their diagnosis in breast cancer. By analyzing, we came to a clearer conclusion that the combination of HRV and serum CEA can assist in the clinical diagnosis of breast cancer at an early stage and improve its early detection rate, thus implementing early intervention and early treatment and reducing the chance of the disease developing to the middle and late stages. There is little research in this area. (p.17-18, lines 322-327).

4. The results section is not well organized. The authors need to state clearly the purpose of each experiment, the observations and conclusions. For example, line 174, the authors examined the correlation between HRV and CEA. The authors have already identified differences of HRV and CEA between patient to control groups. What questions do the authors would like to address by examining the relationship between HRV and CEA and what does the negative correlation between HRV and CEA mean in the context of breast cancer?

Response4: Thank you very much indeed for your comments. We have added the significance of a negative correlation between HRV and CEA in the results section. (p.12, lines 211-216). In the Discussion section, we explain in detail the possible mechanism for the negative correlation between HRV and CEA in breast cancer patients, showing that as breast cancer progresses and cancer cells migrate and metastasize, patients have decreased HRV and increased levels of serum CEA, which is widely present in tumor cell membranes. (p.15-16, lines 265-298).

Spearman's correlation analysis showed that total LF, awake TP, and awake LF were negatively correlated with the CEA index in both groups (P<0.05). Given that CEA has good diagnostic performance for breast cancer as a risk factor and differences in HRV parameters between groups, CEA and HRV parameters may have a joint diagnostic effect on breast cancer. (p.12, lines 211-216).

In addition, we explain in the discussion section that the significant reduction in HRV in breast cancer patients in our experimental results suggests the presence of autonomic dysfunction, and speculate that HRV is of great value in the diagnosis of breast cancer. (p.14-15, lines 247-264). The results of the binary logistic regression analysis suggested that CEA was a risk factor for breast cancer, and the results of the ROC curve analysis suggested that the combination of CEA and HRV was of high value in the diagnosis of breast cancer. (p.16-17, lines 299-313).

5. Joint prediction using combined markers have shown to improve the prediction performance. What computational models did the authors use to the markers (awake TP, awake LF and CEA) are not shown.

Response5: Thanks for your kind suggestions, which is valuable for improving the accuracy of the manuscript. Considering the Reviewer’s suggestion, we have added our approach to deriving the joint prediction model in the article. (p.12, lines 221-223). The logistic regression equation calculates the probability of an outcome based on the values of awake TP, awake LF and CEA, which combine the diagnostic efficacy of awake TP, awake LF and CEA. This union is essentially the C-Statistics for computing logistic regression models. In addition, the data from the logistic regression equation are presented in Supplementary file 6. 

We combined awake TP, awake LF, and CEA separately as a new combined diagnostic model, and then incorporated logistic regression models to derive predictive probabilities, and finally performed ROC curve analysis. (p.12, lines 221-223).

6. How to validate the findings in this manuscript? Have the authors consider independent studies as validation datasets?

Response6: We thank the reviewer for pointing out this issue. We fully agree with the reviewer that we indeed should have applied tests to validate the findings. The results require experiments with a larger sample size to verify their validity. In the future, we will focus more on studying the correlation of tumor markers with autonomic nerve and heart rate variation for further verification. And this is an interesting open issue, and we will continue to consider independent studies as validation datasets. 

We tried our best to improve the manuscript and made some changes in the manuscript. These changes will not influence the content and framework of the paper. And here we did not list the changes but marked with tracked revisions in revised paper. We appreciate for Editors/Reviewers’ warm work earnestly and hope that the correction will meet with approval. Once again, thank you very much for your comments and suggestions.

References:

[1]. Majerova, K., et al., Increased sympathetic modulation in breast cancer survivors determined by measurement of heart rate variability. Sci Rep, 2022. 12(1): p. 14666.

[2]. Wang, Y.Y. and R.H. Sun, [Application of PASS in sample size estimation of non-inferiority, equivalence and superiority design in clinical trials]. Zhonghua Liu Xing Bing Xue Za Zhi, 2016. 37(5): p. 741-4.

[3]. Chen C, Liu HY. Analysis of heart rate variability in patients with different stages of malignant tumors. J Fudan Univ Med Sci. 2009;36(4):413-6

[4]. Tarighati, E., H. Keivan and H. Mahani, A review of prognostic and predictive biomarkers in breast cancer. Clin Exp Med, 2022.

[5]. Wang, W., et al., The diagnostic value of serum tumor markers CEA, CA19-9, CA125, CA15-3, and TPS in metastatic breast cancer. Clin Chim Acta, 2017. 470: p. 51-55.

[6]. Li, X., et al., Clinicopathological and Prognostic Significance of Cancer Antigen 15-3 and Carcinoembryonic Antigen in Breast Cancer: A Meta-Analysis including 12,993 Patients. Dis Markers, 2018. 2018: p. 9863092.

[7]. Kashyap, D., et al., Global Increase in Breast Cancer Incidence: Risk Factors and Preventive Measures. Biomed Res Int, 2022. 2022: p. 9605439.

---

## [Editor Report · Decision Letter 1]

10 Feb 2023

Diagnostic role of heart rate variability in breast cancer and its relationship with peripheral serum carcinoembryonic antigen

PONE-D-22-29053R1

Dear Dr. He,

We’re pleased to inform you that your manuscript has been judged scientifically suitable for publication and will be formally accepted for publication once it meets all outstanding technical requirements.

Kind regards,

Alessandro Rizzo

Academic Editor

PLOS ONE

---

## [Editor Report · Acceptance letter]

29 Mar 2023

PONE-D-22-29053R1 

Diagnostic role of heart rate variability in breast cancer and its relationship with peripheral serum carcinoembryonic antigen 

Dear Dr. He:

I'm pleased to inform you that your manuscript has been deemed suitable for publication in PLOS ONE. Congratulations! Your manuscript is now with our production department. 

Kind regards, 

on behalf of

Dr. Alessandro Rizzo 

Academic Editor

PLOS ONE